# Dimer Rhenium Tetrafluoride with a Triple Bond Re-Re: Structure, Bond Strength

**DOI:** 10.3390/molecules28093665

**Published:** 2023-04-23

**Authors:** Nina I. Giricheva, Natalia V. Tverdova, Valery V. Sliznev, Georgiy V. Girichev

**Affiliations:** 1Nanomaterial Research Institute, Ivanovo State University, Ermak Str. 39, 153025 Ivanovo, Russia; n.i.giricheva@mail.ru; 2Department of Physics, Ivanovo State University of Chemistry and Technology, Sheremetevsky Ave. 7, 153000 Ivanovo, Russia; tverdova@isuct.ru (N.V.T.); sliznev@mail.ru (V.V.S.)

**Keywords:** molecule Re_2_F_8_, Re-Re bond energy, geometric and electronic structure, enthalpy of dimerization, gas electron diffraction, DFT

## Abstract

Based on the data of the gas electron diffraction/mass spectrometry (GED/MS) experiment, the composition of the vapor over rhenium tetrafluoride at T = 471 K was established, and it was found that species of the Re_2_F_8_ is present in the gas phase. The geometric structure of the Re_2_F_8_ molecule corresponding to D_4h_ symmetry was found, and the following geometric parameters of the r_h1_ configuration were determined: r_h1_(Re-Re) = 2.264(5) Å, r_h1_(Re-F) = 1.846(4) Å, α(Re-Re-F) = 99.7(0.2)°, φ(F-Re-Re-F) = 2.4 (3.6)°. Calculations by the self-consistent field in full active space approximation showed that for Re_2_F_8_, the wave function of the ^1^A_1g_ ground electronic state can be described by the single closed-shell determinant. For that reason, the DFT method was used for a structural study of Re_2_X_8_ molecules. The description of the nature of the Re-Re bond was performed in the framework of Atom in Molecules and Natural Bond Orbital analysis. The difference in the experimental values of r(Re-Re) in the free Re_2_F_8_ molecule and the [Re_2_F_8_]^2−^ dianion in the crystal corresponds to the concept of a triple σ^2^π^4^ (Re^IV^-Re^IV^) bond and a quadruple σ^2^π^4^δ^2^ (Re^III^-Re^III^) bond, respectively, which are formed between rhenium atoms due to the interaction of d-atomic orbitals. The enthalpy of dissociation of the Re_2_F_8_ molecular form in two monomers ReF_4_ (Δ_diss_H°(298) = 109.9 kcal/mol) and the bond energies E(Re-Re) and E(Re-X) in the series Re_2_F_8_→Re_2_Cl_8_→Re_2_Br_8_ molecules were estimated. It is shown that the Re-Re bond energy weakly depends on the nature of the halogen, while the symmetry of the Re_2_Br_8_ (D_4d_) geometric configuration differs from the symmetry of the Re_2_F_8_ and Re_2_Cl_8_ (D_4h_) molecules.

## 1. Introduction

There are various rhenium–rhenium bonded complexes, in which rhenium is in different oxidation states. Many experimental and theoretical works have been devoted to the consideration of such compounds [1,2,3,4,5,6]. The most common compounds with a rhenium–rhenium bond are compounds that contain a Re-Re quadruple bond, where the rhenium is in the +3 formal oxidation state (with the d^4^ configuration). This class of compounds stands out not only for its abundance but also for its historical significance.

The quadruple Re^III^-Re^III^ bond in [Re_2_Cl_8_]^2+^ was the first of this kind to be observed; it opened up a new field of study in inorganic chemistry. It is believed that the bond between Re^III^ atoms can be represented as σ^2^π^4^δ^2^ [7,8,9,10,11,12]. The dianions [Re_2_X_8_]^2−^ (where X = F, Cl, Br, I) have a very short Re-Re distance (2.19–2.25 Å) [7,8,9,10,11,12,13,14,15] and there is an eclipsed configuration of two ReF_4_ fragments relative to each other.

As noted in the literature, in contrast to their related compounds (Re^III^, d^4^ configuration) with a lower degree of oxidation, Re^IV^ (d^3^ configuration) complexes are not prone to the formation of metal–metal bonds [1]. Note that all experimental studies of the rhenium complexes structure refer to the condensed state.

In 1993, we performed gas electron diffraction/mass spectrometry (GED/MS) study of rhenium tetrafluoride vapor [16], which contains Re_2_F_8_ dimeric molecules, and determined the geometric structure of this molecular form. It was shown that the model of D_4h_ symmetry with the Re-Re bond best fits the diffraction pattern. That is, the formation of a bond between the Re^IV^ atoms was established. Due to the presence of a short distance r(Re-Re), the assumption of a Re-Re triple bond of σ^2^π^4^ type was made.

The question of what changes occur in the geometric and electronic structure of the [Re_2_X_8_]^2−^ and Re_2_X_8_ complexes with Re^III^-Re^III^ and Re^IV^-Re^IV^ bonds is important for the structural chemistry of compounds with a metal–metal bond.

At present, the methodology for interpreting GED data has changed significantly. The modern GED experiment is a complex study, accompanied by a wide use of the results of quantum chemical calculations, which, along with determining the started geometry for a structural least square analysis of diffraction pattern, make it possible to describe the electronic structure of molecules.

For this, several theoretical methods for obtaining information about electronic properties and bond strength can be used. Population analysis methods such as the Natural Population Analysis (NPA) combined with the Natural Bond Orbital (NBO) method [17] as well as theoretical tools to analyze the topology of the electronic density of molecular systems such as Atom in Molecules (AIM) are among the most popular computational methods for analyzing electronic structures and bonding characters of complexes [18].

In this work, new technical and methodological improvements in the collection and interpretation of GED data were applied (see Section 4). Several density functional theory (DFT) approaches and NPA, NBO and AIM methods have been used to characterize the nature of the Re-Re bond and estimate the bond energies E(Re-Re) and E(Re-X) in the series of Re_2_F_8_ → Re_2_Cl_8_ → Re_2_Br_8_.

Comparison of the experimental structural parameters of Re_2_X_8_ molecules (Re^IV^-Re^IV^ bond) and [Re_2_X_8_]^2−^ dianions (Re^III^-Re^III^ bond) with their calculated analogs makes it possible to recommend the most adequate calculation method for studying the properties of other complexes involving Re^III^ and Re^IV^.

## 2. Results

### 2.1. Analysis of GED/MS Data

The study of Re_2_F_8_ by the GED method could lead to serious errors if performed without mass spectrometric control of the gas phase composition. The rhenium tetrafluoride preparation was obtained by the reduction of ReF_6_ with metallic Re, and therefore, the sample could contain impurities of rhenium compounds in different oxidation states. The dynamics of changes in the mass spectrum at different temperatures confirms this assumption [16]. However, in the temperature range from 450 to 470 K, the relative intensity of ion currents in the mass spectrum over solid ReF_4_ practically does not change, the [ReF_3_]^+^(100), [ReF_4_]^+^(~50), [Re_2_F_7_]^+^(~50) ions, related to Re_2_F_8_, have the highest intensity, while the [ReF_5_]^+^ and [Re_2_F_9_]^+^ ions, which can have a Re_2_F_10_ molecular precursor, are only ~2 and ~1%, respectively. Thus, when analyzing the GED data (T = 471 K), one can use the assumption that the vapor contains a single Re_2_F_8_ molecular form.

### 2.2. Geometric Structure of the Re_2_F_8_ Molecule

Before performing a structural analysis of the GED data, various models of the geometric structure of the Re_2_F_8_ molecule, presented in Figure 1, have been considered. DFT/PBE/RECP-3 theory level was used (the argumentation for choosing this variant and its designation is given in Section 4.2) when optimizing the geometry of each of the structures.

The D_4h_ symmetry model (Figure 1a) has the minimum energy. The optimization of models with non-equivalent Re-F bonds of D_2h_ (Figure 1c) and C_2h_ (Figure 1e) symmetries leads to the same D_4h_ structure. The model D_2h_ with four Re-F_b_ bridging bonds (Figure 1d) is higher in energy than the D_4h_ model (a) by ~100 kcal/mol. The energy of the D_4d_ symmetry model (b) is only ~2.0 kcal/mol higher than that of the D_4h_ model (a). Therefore, the parameters of the two models (a) and (b) were used as starting approximations in least squares (LS) analysis of the GED data.

Appendix A shows the theoretical functions f(r) corresponding to models with the D_4h_ and D_4d_ symmetry of the Re_2_F_8_ molecule. These functions have significant differences in the region of 3–5 Å, indicating the possibility of determining the conformation of the molecule based on GED data.

As a result of the least-squares analysis of the electron diffraction data (details in Section 4.2), it was found that the Re_2_F_8_ molecule in the gas phase has an eclipsed D_4h_ symmetry configuration, the parameters of which are given in Table 1.

### 2.3. GED Data and Frequency of Torsional Vibration of the Re_2_F_8_ Molecule

GED data were used to refine the frequency of torsional vibration ν_tors_, which is less reliably determined in quantum chemical calculations, and more than other frequencies depends on the calculation theory level. The value of the torsion force constant significantly affects the amplitudes of vibrations of all F…F terms. Therefore, there is a fundamental possibility of determining ν_tors_ from GED data, especially since this vibration is characteristic and does not depend on other internal coordinates.

We have made an attempt to estimate ν_tors_ from its dependence on the disagreement factor between the experimental and theoretical functions of the reduced molecular scattering intensity R_f_ = f(ν_tors_) (Appendix A, details in Section 4.2). It turned out that the value of ν_tors_ = 30 cm^−1^ corresponds best to the GED experiment.

Figure 2 and Figure 3 show the experimental and theoretical functions sM(s) and f(r), which indicate that the geometric and vibrational characteristics of Re_2_F_8_ molecule with D_4h_ symmetry (Table 1) are in agreement with the diffraction data.

### 2.4. Comparison of the Results of Two Methodology Approaches to LS-Analysis

Table 1 shows the r_h1_ obtained in this work and r_α_ [16] structural parameters of the Re_2_F_8_ molecule. These parameters possess similar physical meaning but differ in the method of calculating vibrational corrections to the values of internuclear distances and root mean square vibrational amplitudes l (Table 1). Parameters of both r_h1_ and r_α_ structures are approximations to the r_e_ parameters of the equilibrium configuration of the molecule. In [16], the harmonic approximation and linearized vibrational coordinates were used to calculate the vibrational amplitudes and vibrational corrections to the averaged internuclear distances. In this work, curvilinear vibrational coordinates were used. The latter technique allows obtaining r_h1_ parameters that are closer in physical meaning to the r_e_ parameters of the equilibrium configuration than r_α_ parameters [19,20,21].

It can be seen that despite the different physical meaning, the r_h1_ and r_α_ for the Re-Re bond are close. As for Re-F bond distance, a new obtained value is significantly close to the value predicted by PBE calculations (Table 1). It should be noted that the use of new methods of photometric experiment and a new approach to structural analysis (details in Section 4.2) allowed us to reduce the value of the disagreement factor R_f_ significantly (from 7% to 4.7%)

Thus, the previous GED study [16] of the Re_2_F_8_ molecule performed with an estimated force field gave the right conclusion concerning the symmetry of equilibrium configuration of this molecule without relying on quantum chemical calculations. However, the latter make it possible to significantly expand the understanding of the fine details of the geometric and electronic structure of molecules.

## 3. Discussion

### 3.1. The Nature of the Re-Re Chemical Bond in the Re_2_F_8_ Molecule

Saito [14] described in detail the Re^III^-Re^III^ bond nature in the [Re_2_Cl_8_]^2−^ dianion, which has a four multiplicity of σ^2^π^4^δ^2^ type. As noted in [14], one should expect that in a neutral Re_2_F_8_ molecule, the Re^IV^-Re^IV^ bond will have the character of a σ^2^π^4^ triple bond. Our study confirms this assumption and develops ideas about the chemical bond in Re_2_F_8_.

The diagram of eight border canonical molecular orbitals (MO) and their view are shown in Figure 4. The d_z2_, d_xz_, d_yz_ or d_xy_-AOs of two Re atoms make the main contribution to all noted MOs.

The d_x2-y2_ AOs are mainly involved in the formation of σ(Re-F) bonding and σ*(Re-F) antibonding MOs related to Re-F bonds of the Re_2_F_8_ molecule. They are in energy lower and higher MOs, as presented in Figure 4. The σ(Re-F) NBO and σ*(Re-F) NBO are shown in Figure 5.

If we neglect the contributions from the p-AO of fluorine atoms, then the MOs presented in Figure 4 can be classified as follows: the a_1g_ symmetry orbital represents the occupied bonding σ(Re-Re) MO built from d_z2_ AOs; two occupied orbitals of e_u_ symmetry represent degenerate bonding MOs π_x_(Re-Re) and π_y_(Re-Re) built from d_xz_ or d_yz_ AOs; the unoccupied b_2g_ orbital refers to the bonding δ(Re-Re) MO, consisting of d_xy_ AOs; orbitals of symmetry b_1u_, e_g_, and a_2u_ refer to antibonding δ*(Re-Re), π_x_*(Re-Re) = π_y_*(Re-Re) and σ*(Re-Re) MOs, respectively. The electronic configuration a_1g_^2^e_u_^4^ corresponds to a σ^2^π^4^ triple bond between rhenium atoms.

The results of AIM and NBO analysis (Table 2), despite the difference in approaches to the analysis of the electronic structure of the molecule, demonstrate agreement in the values of bond orders (δ_AIM_/P_NBO_), which are more than two for the Re-Re bond and close to 1 for the Re-F bonds. The bond ellipticity (ε) of the Re-Re is equal to 0, which confirms the σ^2^π^4^ character of this bond. The net charges on the Re and F atoms indicate a noticeable ionic component of the Re-F bond.

### 3.2. Free Re_2_F_8_ Molecule in the Gas Phase and [Re_2_F_8_]^2−^ Dianion in Crystals; the Re^IV^-Re^IV^ Bond vs. the Re^III^-Re^III^ Bond

In [Re_2_X_8_]^2−^ dianions with D_4h_ symmetry, the molecular orbital δ(b_2g_) of the Re-Re bond becomes the highest occupied molecular orbital (HOMO), the δ*(b_1u_) orbital is the lowest unoccupied molecular orbital (LUMO) (Figure 4), and the difference between the energies of the frontier orbitals is much smaller than between frontier orbitals in Re_2_X_8_ molecules (X = Hal). Therefore, the methods of quantum chemical calculations for the Re_2_X_8_ molecule and the [Re_2_X_8_]^2−^ dianion are fundamentally different ([14] and Section 4.1).

Nevertheless, it is possible to compare the geometric and vibrational parameters of the molecules and the dianions calculated using the DFT because this approximation reliably predicts the structure of the mentioned species (Appendix A).

Table 3 shows the experimental internuclear distances and the calculated frequencies of ν(Re-Re) stretching vibration and ν_tors_ torsion vibration as well as NPA charges on atoms in two species Re_2_F_8_ and [Re_2_F_8_]^2−^.

Both species have an eclipsed D_4h_ geometric configuration with ^1^A_1g_ ground state symmetry, the electronic configurations of which correspond to the triple σ^2^π^4^ and quadruple σ^2^π^4^δ^2^ Re-Re bonds, respectively. Those Re-Re bond should become shorter, and the frequency of the ν(Re-Re) stretching vibration should be higher when going from Re_2_F_8_ to [Re_2_F_8_]^2−^.

The NPA charges on the F atoms in Re_2_F_8_ and [Re_2_F_8_]^2−^ indicate an increase in the energy of steric repulsion between them in the dianion, which leads to the lengthening of all F…F distances and an increase in the bond angle FReRe. At the same time, a larger difference between the charges on the Re and F atoms can be one of the factors for the shortening of the Re-F distance in the Re_2_F_8_ molecule.

In addition, both in the molecule and in the dianion, there are donor–acceptor interactions between the bonding σ(Re1-F) NBO of one ReF_4_ fragment and the antibonding σ*(Re2-F) NBO of another ReF_4_ fragment (Figure 5).

As a result of such interactions, the electron density is transferring from the bonding NBO to the antibonding one. This circumstance leads to a weakening of both the Re1-F bond and the Re2-F bond. Since the energy of this interaction in the dianion is almost two times greater than in the molecule, the distance r (Re-F) in the dianion is longer, and the frequency of the stretching vibration ν (Re-F) is lower.

This is evidenced by the bond orders calculated according to the Weiberg scheme [17]: P (Re-Re) = 2.24 and 3.10, P (Re-F) = 0.81 and 0.59 for Re_2_F_8_ and [Re_2_F_8_]^2−^, respectively.

The presence of the δ (Re-Re) bond, as well as the presence of such strong donor–acceptor interactions that prevent internal rotation, can explain the greater value of the torsional vibration frequency in the dianion.

### 3.3. Structural Changes in the Series Re_2_F_8_ → Re_2_Cl_8_ → Re_2_Br_8_

This section presents the results of calculations for Re_2_Cl_8_ and Re_2_Br_8_ molecules. Experimental data on their structure are absent in the literature. The same variant of the DFT method was used, which led to better agreement between the calculated and experimental parameters of the Re_2_F_8_ molecule (Section 4.2). Moreover, the calculated structural characteristics of the [Re_2_X_8_]^2−^ dianions of these compounds coincided within the error limit with the experimental data found by X-ray diffraction analysis [9,10,11,12,13] (Appendix A).

Some molecular characteristics of the three Re_2_X_8_ molecules are given in Table 4, allowing to trace the influence of the nature of the atom halogen X on their structure. Thus, r (Re-Re) increases by ~0.08 Å during the transition Re_2_F_8_ → Re_2_Cl_8_ → Re_2_Br_8_ (D_4h_), while in the dianion series, this increase is noticeably weaker (~0.04 Å, Appendix A), which is due to the stronger and shorter Re-Re bond in dianions.

With the lengthening of the Re-Re bond, the frequency of the ν (Re-Re) stretching vibration decreases as expected (Table 4). The Re-Re-X and X-Re-X bond angles change by no more than 3°, and they are close to 90°, which indicates a significant contribution of 5d_x2-y2_ AO of rhenium atoms to the formation of Re-X bonds.

The distances r (X…X) between halogen atoms in the cis position with respect to each other are less than the sum of their van der Waals radii, and the difference between r (X…X) and Σr_VdV_ (X) increases when going from F to Br. This leads to an increase in steric repulsion between ReX_4_ fragments in Re_2_X_8_ molecules.

### 3.4. D_4h_ or D_4d_ Symmetry of Re_2_X_8_ Molecules?

It was noted in [14] that in contrast to [Re_2_X_8_]^2−^ dianions with a quadruple σ^2^π^4^δ^2^ Re-Re bond, in the neutral form Re_2_X_8_, the free internal rotation of ReX_4_ groups around the σ^2^π^4^ triple bond, which has axial symmetry, is possible.

It is assumed [14] that when the symmetry of the molecule is lowered from D_4h_ to D_4d_, the steric repulsion energy between X atoms of two ReX_4_ groups will decrease, which will make the D_4d_ configuration of Re_2_X_8_ molecules more energetically favorable than D_4h_.

However, as noted in Section 2.2, the GED data indicate the D_4h_ symmetry of the geometric configuration of the Re_2_F_8_ molecule. To confirm this conclusion, we scanned the potential energy surface along the torsion coordinate F-Re-Re-F. The calculation of the total energy at each point in Figure 6 was carried out at a fixed value of the dihedral angle φ (F-Re-Re-F) by optimizing all other geometric parameters while maintaining the D_4_ symmetry.

The analysis of molecular orbitals carried out for all points on the internal rotation function (Figure 6) showed that the shape and composition of the molecular orbitals responsible for the formation of the Re-Re triple bond remain practically unchanged. Apparently, the constancy of the electron density distribution in the region of the Re-Re bond is the reason for the low barrier of internal rotation with respect to the triple bond.

The PFIR (Figure 6) indicates that the D_4h_ symmetry model is a stable Re_2_F_8_ structure, and the geometry of D_4d_ symmetry corresponds to a first-order saddle point, where the imaginary frequency refers to the torsional vibration. It can be seen that the barrier of internal rotation is ~2 kcal/mol, and it significantly exceeds the thermal energy RT (~0.94 kcal/mol), which corresponds to the temperature of the GED experiment and the change in the torsion angle ϕ (F-Re-Re-F) = 0 ± 21°. In this case, the vibration amplitude of the F5…F10 term is equal to 0.325 Å, and it corresponds to the experimental value from GED (Table 1).

For the Re_2_Cl_8_ molecule, the barrier value decreases (to 1.1 kcal/mol), as well as the value of ν_tors_ (Re-Re), which indicates a greater structural non-rigidity of Re_2_Cl_8_ compared to the Re_2_F_8_ molecule.

At last, for the Re_2_Br_8_ molecule, the D_4d_ symmetry configuration becomes more stable compared to D_4h_ (Table 4), which becomes the first-order saddle point on the potential energy surface (PES). Such a change in the energy difference E(D_4d_) − E(D_4h_) in the series Re_2_F_8_ → Re_2_Cl_8_ → Re_2_Br_8_ corresponds to an increase in the steric repulsion energy between halogen atoms because of small changes in the Re-Re bond length and the Re-Re-X bond angle and an increase in the effective size of halogens.

### 3.5. Re_2_F_8_ Dissociation Enthalpy and Re-Re and Re-X Bond Energy in Re_2_X_8_ Molecules

The enthalpy of dissociation of Re_2_F_8_ was estimated in [22] basing the experimental values of the standard enthalpies of formation of gaseous and solid rhenium tetrafluoride and the enthalpy of Re_2_F_8_ sublimation. The value Δ_diss_H° (298) was estimated as 112 ± 16 kcal/mol. To verify this value, we have considered the gas-phase dissociation reaction
Re_2_F_8_ → 2∙ReF_4_
using the PBE/RECP-3 method (see Section 4.2).

For this, geometric optimization of the ReF_4_ molecule (q = 0) with a multiplicity (M) of electronic state equal to 4 was performed, which implies the presence of three unpaired electrons. According to the crystal field theory, the initial ReF_4_ tetrahedral configuration must undergo Jahn–Teller distortions and reduce the symmetry to D_2d_. The calculation results confirm this.

NBO analysis shows that three unpaired electrons are located on 5d_xy_, 5d_x2-y2_, and 5d_z2_ AOs, resulting in a distorted tetrahedral structure extended along the x-axis (Figure 7). 

The energy of the Re_2_F_8_ dissociation reaction was calculated by Equation (1) as the difference between the electronic energies of the Re_2_F_8_ and ReF_4_ molecules:ΔE_diss_ = 2·E[ReF_4_ (D_2d_)] − E[Re_2_F_8_ (D_4h_)](1)
and amounted to ΔE_diss_ = 111.68 kcal/mol.

The enthalpy and Gibbs free energy of dissociation were calculated in a similar way:Δ_diss_H° (298) = 109.9 kcal/mol
Δ_diss_G° (298) = 93.2 kcal/mol

Note that the calculated value of Δ_diss_H (298) coincides with the value [22] estimated from experimental thermochemical data (112 ± 16 kcal/mol).

For Re_2_F_8_, Re_2_Cl_8_, and Re_2_Br_8_ molecules, the Re-Re bond energy was estimated taking into account the basis set superposition error (BSSE) [23,24]. It was assumed that the original Re_2_X_8_ molecule consists of two ReX_4_ fragments (q = 0, M = 4). The BSSE correction turned out to be small (~1 kcal/mol).

The Re-X bond energy was estimated according to Equation (2):E(Re-X) = E(Re_2_X_7_∙) + E(F) − E(Re_2_X_8_)(2)
where E(Re_2_X_7_∙) is the electronic energy of the radical Re_2_X_7_ ∙ (M = 2) with its geometry in the Re_2_X_8_ molecule, and E(F∙) is the electronic energy of the F atom (M = 2).

The Re-Re and Re-X bond energies in Re_2_F_8_, Re_2_Cl_8_ and Re_2_Br_8_ molecules are given in Table 4. As can be seen, the energies E(Re-Re) change slightly when replacing halogens. There is a close to linear correlation between the calculated Re-X bond energy and the internuclear distance r(Re-X) (Figure 8.).

## 4. Materials and Methods

### 4.1. Details of Calculations

Most of the quantum chemical calculations in this work were performed by the DFT method. The argument for using DFT was the results of calculations of the total energies of singlet and triplet electronic states by the self-consistent field in the full active space (CASSCF) method, which was followed by taking into account the dynamic correlation of electrons in the framework of the multiconfiguration, quasi-degenerate second-order perturbation theory (MCQDPT2) [25,26] for the Re_2_F_8_ molecule using the FireFly 8.2 program [27].

In this case, geometric parameters obtained by optimization in the DFT/PBE0 approximation were used.

The active space of the CASSCF method included 6 electrons and 10 molecular orbitals, consisting mainly of a combination of the corresponding components of the 5d orbitals of two rhenium atoms. Calculations in the CASSCF approximation performed for the singlet states of the equilibrium D_4h_ symmetry configuration of the Re_2_F_8_ molecule showed that the weight of the leading Slater determinant in the wave function of the ground singlet electronic state (^1^A_1g_) is 86%.

The relative energy of the nearest excited singlet state is 127 kcal/mol. Calculations of the total energies of triplet electronic states performed in the CASSCF and MCQDPT2 approximations showed that the lower triplet term (^3^E_u_) lies above the ground singlet electronic state by 57 and 23 kcal/mol, respectively.

The leading determinant in the wave function of the ground electronic state ^1^A_1g_ corresponds to the closed-shell electron configuration, which was implemented in DFT calculations.

DFT calculations were performed with three functional/basis combinations and different pseudopotentials of the Re atom. The calculated geometric parameters of the Re_2_F_8_ molecule, obtained in three versions, are compared with the experimental data in Table 5.

Calculations in the PBE0 approximation [28] were performed using the FireFly 8.2 software [27]. The quasi-relativistic effective pseudopotential (qRECP) and correlation consistent basis set [29] was used for the Re atom. For the fluorine atom, the Sapporo basis (SPKrATZP) [30,31] supplemented with diffuse functions was applied (referred to as PBE0/qRECP-1).

Calculations with the B3LYP [32,33,34] and PBE [35] functionals were performed using the Gaussian09 program [36].

In the DFT/B3LYP calculation, the quasi-relativistic effective core potentials [29] and the basis set [37] on the Re atom was used. Fluorine atoms were described using the correlation-consistent valence-three-exponential cc-pVTZ basis set [38] (referred to as B3LYP/qRECP-2).

In the DFT/PBE approximation, the relativistic effective pseudopotentials (RECP) and the corresponding aug-cc-pVTZ-PP basis set [39] was used to describe the rhenium atom. The basis set aug-cc-pVTZ was applied to describe fluorine [38,40], chlorine [41], and bromine [42] atoms (referred to as PBE/RECP-3).

Effective core potentials and basis sets in the Gaussian program format were adopted from the Basis Set Exchange library (BSE) [43,44,45].

For the ground electronic states, the topological analysis of electron density distribution function *ρ*(r) for the molecules under study was carried out using AIMAll Professional software [18]. The NBO 5G program [17], implemented for natural orbital analysis in FireFly 8.2 [27], was used to obtain the net atomic charges, and to study the effect of hyperconjugation on the structure. Visualization of the geometrical structures and orbitals was performed by the ChemCraft program [46].

### 4.2. Features of Structural Analysis of GED/MS Data

The gas phase electron diffraction patterns and mass spectra were recorded simultaneously using the technique described in ref. [47,48]. The conditions of the synchronous gas-phase electron diffraction/mass spectrometric (GED/MS) experiments are shown in Appendix A. The optical densities of exposed photoplates were recorded for two nozzle-to-plate distances: *L*_1_ = 598 mm and *L*_2_ = 338 mm. Seven electron diffraction patterns of the substance and two electron diffraction patterns of the ZnO crystal standard were recorded for each nozzle-to-plate distance.

In Ref. [16], electron diffraction patterns were scanned by their diameter. As a result, the diffraction pattern for each photographic plate was represented by a set of 301 points. In this case, the complete statistical sample for all electron diffraction patterns was 4214 points.

In contrast to [16], we increased significantly the sample of scanning results and precision of optical density measurements. To achieve this goal, we used a modified MD-100 microdensitometer [49] and measured the optical density on each plate by scanning an area of 10 × 130 mm with a step of 0.1 mm along 33 equidistant scanning lines. As a result of new microphotometric measurements, the statistical sample of primary experimental data was significantly increased in comparison with [16] and amounted to more than 500,000 points. 

The geometric model of the Re_2_F_8_ molecule was specified by four independent parameters: two internuclear distances Re-Re, Re-F, bond angle FReRe and dihedral angle FreReF. The values of dependent internuclear distances were determined within the framework of the r_h1_ structure. The independent parameters together with five groups of root-mean-square vibration amplitudes were varied during the least-squares analysis of GED data using the modified KCED-35 program [50].

The VibModule program [51] was applied to calculate the vibrational corrections Δ*r* = *r_h_*_1_ − *r_a_* and the starting values of root-mean-square vibration amplitudes of Re_2_F_8_ at the temperature of the GED experiment using the harmonic approximation and taking into account the non-linear interrelation between internal and Cartesian vibrational coordinates.

The DFT method with three different functionals and basis sets (see Section 4.1) was used to calculate the starting molecular characteristics of the Re_2_F_8_ molecule (Table 5). The LS analysis of GED data showed that three variants of starting parameters lead to identical geometric and vibrational parameters of the experimental structure, which indicates the stability of the solution of the GED inverse task.

Table 5 shows the geometrical parameters of the D_4h_ symmetry structure, the frequency of stretching vibrations ν(Re-Re) and stretching vibrations ν(Re-F), as well as the torsional vibration ν_tors_(Re-Re), which are calculated in three functionals of DFT for the Re_2_F_8_ molecule.

Comparison of the calculated parameters with the experimental ones (Table 5) shows that the PBE/RECP-3 combination has an advantage over other calculation variants and may be used to estimate the geometric structure of the Re_2_X_8_ molecules as well as of the [Re_2_X_8_]^2−^ dianions.

Appendix A show the vibrational frequency values of the molecule Re_2_F_8_ obtained in three variants using DFT. Differences in the values of the Re-F stretching vibration frequencies (590–750 cm^−1^) and the F-Re-F and Re-Re-F bending vibrations (125–750 cm^−1^) lead to changes in vibrational amplitudes of terms F…F, which do not exceed the error of their determination in the GED experiment, when using the same value of the torsion vibration frequency. At the same time, as noted in Section 2.3, the value of the torsion force constant significantly affects the vibrational amplitudes of all terms F…F.

To determine the value of the ν_tors_, the different torsion force constants were used for the calculation of starting values of the vibrational amplitudes by the VibModule program [51], which were then refined in the LS analysis of the GED data (Appendix A). It turned out that at ν_tors_ = 30 cm^−1^, the value of the disagreement factor R_f_ between the experimental and theoretical functions sM(s) acquires a minimum value, and the calculated amplitude F5…F10 coincides with the experimental one.

As a result of the applied and described above improvements in the quality of photometric measurements and the use of quantum chemical calculations for obtaining the force field of a molecule and an improved method for calculating the vibrational characteristics of a molecule [19,20,21], it was possible to achieve a significantly better agreement between the model scattering intensity function and experiment (R = 4.7% versus 7.0% in [16]).

## 5. Conclusions

The reinterpretation of the diffraction data [16] for gaseous rhenium tetrafluoride has been carried out at a higher methodological level. In the temperature range from 450 to 470 K, the dominant species of the vapor is Re_2_F_8_. The Re_2_F_8_ molecule possesses a geometric structure of D_4h_ symmetry. Based on the GED data, the frequency of torsional vibrations relative to the Re-Re bond, which is associated with the structural non-rigidity of the molecule, was estimated.

The electronic structure of Re_2_F_8_ was determined in the CASSCF and MCQDPT2 approximations, and it was shown that the ^1^A_1g_ ground electronic state can be described by a single-reference wave function. This circumstance indicates the possibility of using the DFT method for extended analysis of the geometric and electronic structure of molecules in the series Re_2_F_8_ → Re_2_Cl_8_ → Re_2_Br_8_.

The potential function of the internal rotation of ReF_4_ fragments relative to the Re-Re bond of the Re_2_F_8_ molecule has been calculated. Structural changes in the series Re_2_F_8_ → Re_2_Cl_8_ → Re_2_Br_8_ free molecules were considered. The transition from fluoride to chloride leads to a decrease in the barrier V = E(D_4d_) − E(D_4h_), and for Re_2_Br_8_, the D_4h_ symmetry structure becomes a saddle point on the PES. This fact does not contradict the concept of an increase in the steric repulsion between ReX_4_ fragments with an increase in the effective size of the X halogen atom. The heat of dissociation of the Re_2_F_8_ species (Δ_diss_H^°^(298) = 109.9 kcal/mol). The bond energies E(Re-Re) and E(Re-X) in the series Re_2_F_8_ → Re_2_Cl_8_ → Re_2_Br_8_ molecules were estimated.

The results of the NBO analysis and QTAIM, despite the difference in approaches to the analysis of the electronic structure of the molecule Re_2_F_8_, demonstrate agreement in the values of bond orders, which are more than 2 for the Re-Re bond and close to 1 for the Re-F bond. The ellipticity of the Re-Re bond is equal to 0 and confirms the σ^2^π^4^ character of this bond. The net charges on the Re and F atoms indicate a noticeable ionic component of the Re-F bond. Structural features of free Re_2_F_8_ molecule in the gas phase and [Re_2_F_8_]^2−^ dianion in crystals are considered.

There is a shortening of the Re-Re bond and an increase in the ReReF bond angle upon passing from Re_2_F_8_ to [Re_2_F_8_]^2−^, and this fact corresponds to the concept of a triple σ^2^π^4^ (Re^IV^-Re^IV^) bond and a quadruple σ^2^π^4^δ^2^ (Re^III^-Re^III^) bond, which are formed between rhenium atoms due to the interaction of d AOs in individual molecules and in ion fragments of crystal, correspondingly.

## Figures and Tables

**Figure 1 molecules-28-03665-f001:**
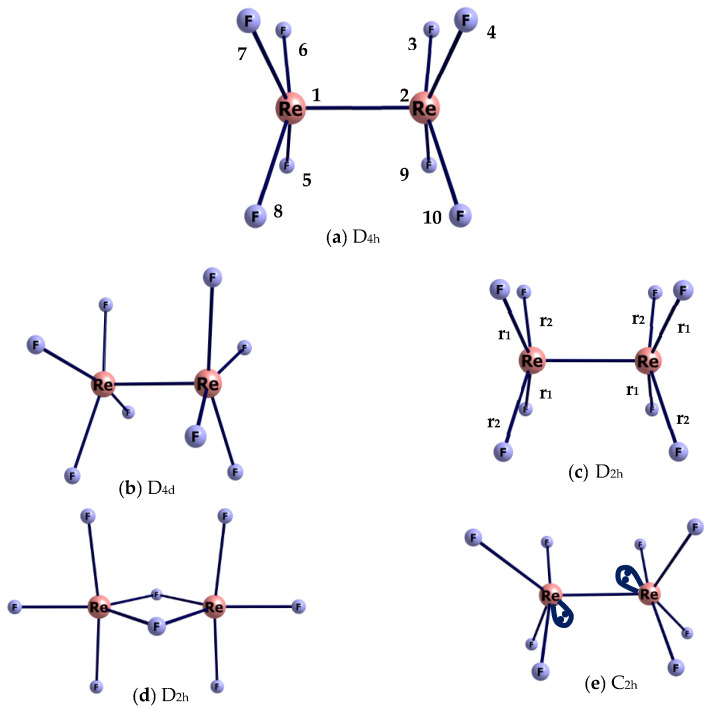
Optimized structures of Re_2_F_8_ molecule: (**a**) model D_4h_ with an eclipsed conformation; (**b**) model D_4d_ with a staggered conformation; (**c**) model D_2h_ with non-equivalent Re-F bonds; (**d**) model D_2h_ with four Re-F_br_ bridging bonds 4; (**e**) model with an ordinary Re-Re bond of C_2h_ symmetry.

**Figure 2 molecules-28-03665-f002:**
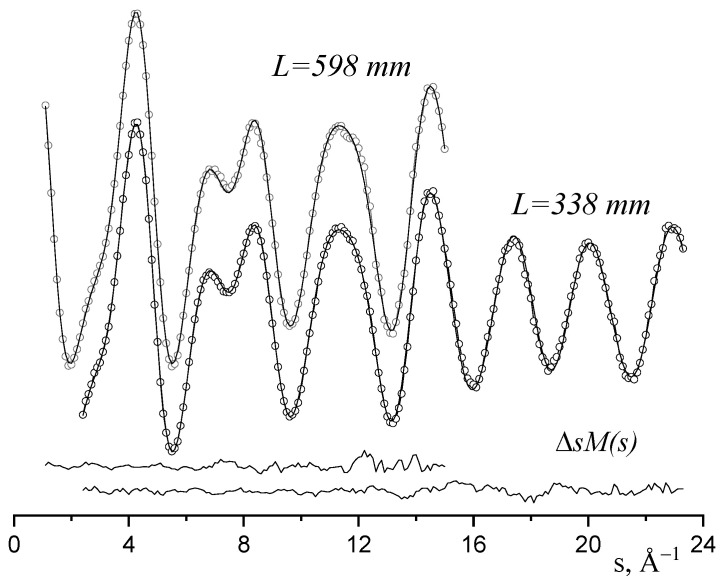
The experimental (circles) reduced molecular scattering intensity sM(s) of the ReF_4_ vapor at T = 471 K and theoretical (full lines) one, corresponding to the D_4h_ model of Re_2_F_8_ molecules, and difference curves ΔsM(s).

**Figure 3 molecules-28-03665-f003:**
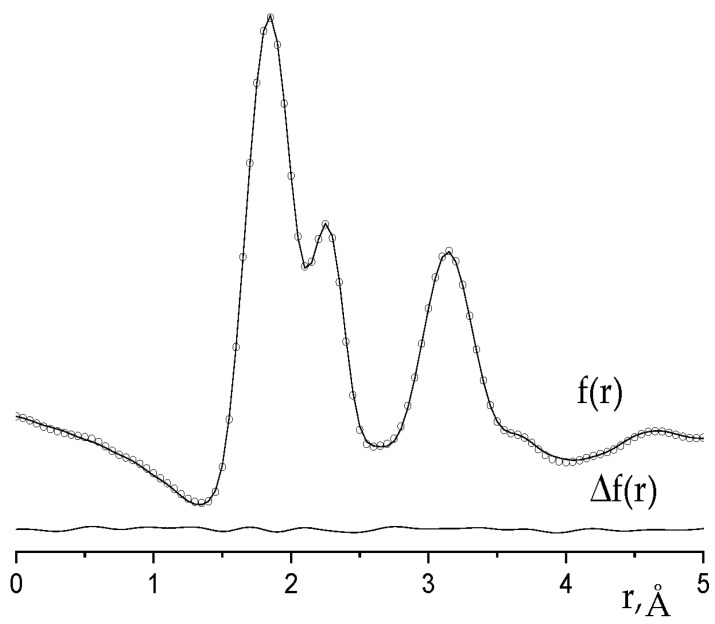
Radial distribution functions f(r) (experimental–circles, theoretical—full line) and difference curve Δf(r).

**Figure 4 molecules-28-03665-f004:**
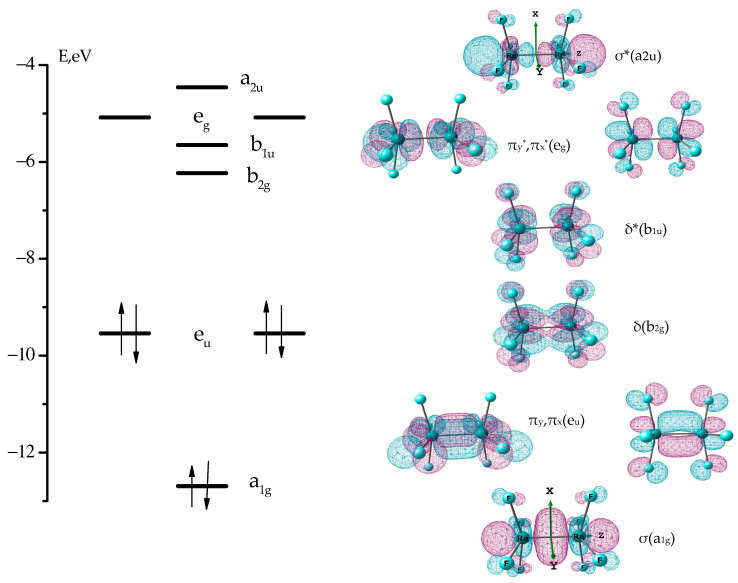
Diagram and view of the MOs, the main contribution to which is made by d_z2_, d_xz_, d_yz_, and d_xy_-AO of two Re atoms.

**Figure 5 molecules-28-03665-f005:**
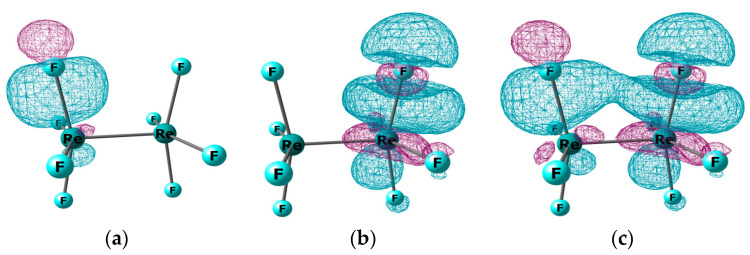
(**a**) Donor σ(Re1-F) NBO, (**b**) Acceptor σ*(Re2-F) NBO, (**c**) result of their interaction. NBOs are built from sp^2.7^d^2.9^ hybrid Re orbitals and sp^2.5^ hybrid F orbitals.

**Figure 6 molecules-28-03665-f006:**
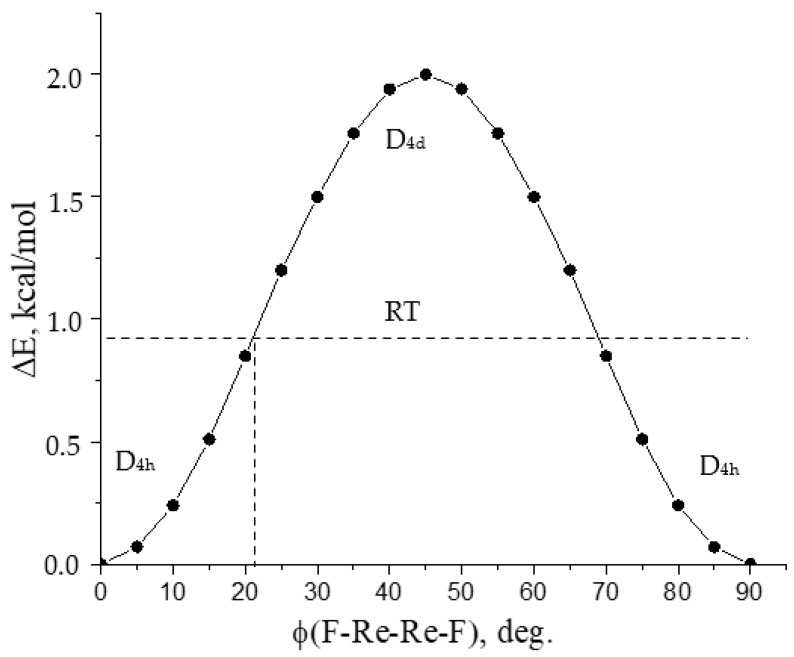
Potential function of the internal rotation (PFIR) of ReF_4_ fragments about the Re-Re bond of the Re_2_F_8_ molecule.

**Figure 7 molecules-28-03665-f007:**
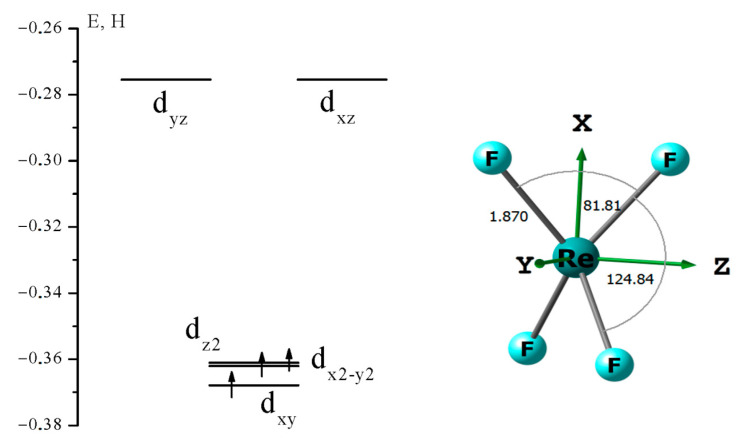
Diagram of rhenium 5d AOs and geometric structure of ReF_4_.

**Figure 8 molecules-28-03665-f008:**
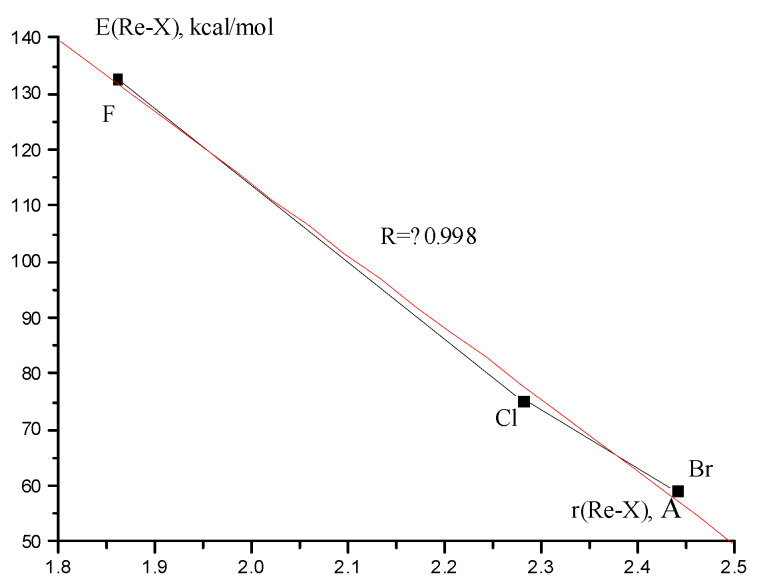
Correlation between Re-X bond energy and internuclear distance r(Re-X).

**Table 1 molecules-28-03665-t001:** Structural parameters of r_h1_ and r_α_ structures (in Å) of the Re_2_F_8_ molecule according to the data of GED experiments and PBE/RECP-3 calculation.

Parameters	PBE	r_h1_	l_calc_	l_exp_	r_α_ [16]	l_calc_ [16]	l_exp_ [16]
Re1-Re2	2.259	2.264 (5)	0.038	0.046 (2)	2.269 (5)	0.035	0.035 (2)
Re1-F6	1.861	1.846 (4)	0.044	0.045 (4)	1.830 (4)	0.042	0.041 (5)
F5…F8	2.592	2.573 (6)	0.115	0.123 (2)		0.117	0.122 (16)
F5…F9	2.901	2.887 (9)	0.240	0.238 (4)		0.199	0.192 (8)
Re1…F9	3.165	3.153 (7)	0.116	0.114 (4)	3.123 (10)	0.108	0.100 (8)
F5…F7	3.665	3.639 (8)	0.071	0.074 (14)		0.064	0.085 (30)
F5…F10	3.890	3.830 (48)	0.314	0.317 (14)		0.283 ÷ 0.452	0.33 (8)
F5…F4	4.674	4.643 (10)	0.115	0.161 (34)		0.111	0.18 (8)
F5Re1Re2	99.9	99.7 (2)			98.7 (7)		
F4ReReF7		2.4 (3.6)					
R_f_, %		4.7			7.0		

**Table 2 molecules-28-03665-t002:** AIM/NBO characteristics of Re-F and Re-Re bonds in Re_2_F_8_ by PBE0/qRECP-1 data ^a^.

**q_AIM_/q_NPA_ (Re)**	2.36/1.57	
**q_AIM_/q_NPA_ (F)**	−0.59/−0.39	
**bond**	Re-F	Re-Re
**ε**	0.074	0.000
**δ_AIM_/P_NBO_**	0.83/0.81	2.28/2.24

^a^ q—a net charge of an atom, e; ε—a bond ellipticity; δ—a electron delocalization index, e, P_NBO_—Wiberg bond index.

**Table 3 molecules-28-03665-t003:** Selected parameters of the Re_2_F_8_ molecule and the [Re_2_F_8_]^2−^ dianion.

	r (Re-Re), Å	r (Re-F), Å	FReRe, °	ν (Re-Re) cm^−1^	ν_tors_ cm^−1^	q (Re)	q (F)
Re_2_F_8_ GED, D_4h_	2.264 (5)	1.846 (4)	99.7 (2)	345 ^a^	30	1.57 ^a^	−0.39 ^a^
[Re_2_F_8_]^2−^ X-ray, D_4h_	2.188 (3) [15] 2.20 [12]	1.95 [12]	104.5 ^a^	353 ^a^	69 ^a^	1.23 ^a^	−0.56 ^a^

^a^ calculated parameters at PBE/RECP-3 theory level.

**Table 4 molecules-28-03665-t004:** Selected molecular characteristics of Re_2_F_8_, Re_2_Cl_8_ and Re_2_Br_8_
^a^.

Molecular Characteristics	Re_2_F_8_ D_4h_	Re_2_Cl_8_ D_4h_	Re_2_Br_8_ D_4h_/D_4d_
r (Re-Re), Å	2.259	2.317	2.336/2.300
r (Re-X), Å	1.860	2.280	2.441/2.445
Re-Re-X, °	99.9	101.6	102.6/102.0
X-Re-X, °	88.3	87.7	87.3/87.5
r (X…X), Å	2.901	3.234	3.402/3.786
Σr_VdV_ (X), Å ^b^	2.94	3.5	3.9
ν (Re-Re), cm^−1^	338	268	261/275
ν_tor_ (Re-Re), cm^−1^	42.6	20.2	12.5i/9.1
E (D_4d_)-E (D_4h_), kcal/mol	2.0	1.1	−1.0
E (Re-Re), kcal/mol	106.1	101.1	99.8
E (Re-X), kcal/mol	132.7	75.4	59.1

^a^ PBE/RECP-3, ^b^ Σr_VdV_ (X)—sum of van der Waals radii.

**Table 5 molecules-28-03665-t005:** Results of three variants of DFT calculations and experimental data for the Re_2_F_8_ molecule.

	r_e_ (Re-Re) Å	r_e_ (Re-F) Å	α_e_ (ReReF) °	ν (Re-Re) cm^−1^	ν_tors_ (Re-Re) cm^−1^	ν (Re-F) ^a^ cm^−1^
PBE0/qRECP-1	2.226	1.843	100.1	370	40	617–749
B3LYP/qRECP-2	2.288	1.871	99.7	345	10	594–710
PBE/RECP-3	2.259	1.860	99.9	338	43	597–704
GED	2.264 (5)	1.846 (4)	99.8 (0.2)		30	

^a^ the range of eight stretching Re-F vibration frequencies (Appendix A).

## Data Availability

The data presented in this study are available on request from the corresponding author.

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
