# Peer review of "Dimer Rhenium Tetrafluoride with a Triple Bond Re-Re: Structure, Bond Strength"

_molecules, 2023, doi:10.3390/molecules28093665_

Round 1

Reviewer 1 Report

I do not see any strong objections against the manuscript. The paper is well done and experimental data are well confirmed by the quantum calculations. Taking into account that the quantum chemical calculations for cluster chemistry is rare and quite difficult the paper is worthily to be published.  

Author Response

Dear Reviewer,
We are grateful to you for reading our article and for your approval of its publication in the Special Issue "DFT Quantum Chemical Calculation of Metal Clusters" of the journal Molecules.
Nina I. Giricheva, Natalia V. Tverdova, Valery V. Sliznev, Georgiy V. Girichev

Reviewer 2 Report

In the work entitled “Dimer Rhenium Tetrafluoride with a Triple Bond Re-Re: Structure, Bond Strength”, Giricheva et al. performed DFT calculations on the structures of Re2X8 molecules, where X = F, Cl, Br.

Sometimes, the authors say Re2X82-, sometimes Re2X8, what exactly the charges (-2 or 0) of Re2X8 are in their study? And what is the charge in the dissociated ReX4?

The authors did not make their work clear to the readers. Why they chose this topic? What impacts are their conclusions? The writing of Abstract and the main text is required to be improved and polished.

It seems the topic of the work is very old , since all the references cited in the work are more than ten years ago. Recent related literature are recommended to be cited and discussed.

Some abbreviations appeared in the manuscript for the first time should give the full expression at the same time, such as AO, DFT, QTAIM, LS, HOMO, LUMO, NBO, etc.

PBE is a type of GGA functional, while PBEPBE is the keyword of PBE functional in Gaussian and some other software.

Authors might provide what are ”RECP-2” and (F)ECP-3 in their computations.

There are some grammar issues. For example, “In this work the plates with diffraction patterns of rhenium tetrafluoride [10] were undergo the new photometric experiment using improved equipment and used state-of-art methodics of structural analysis (see Section 4)” in the last paragraph of Introduction, if authors meant to use undergo, it should be underwent.

In the first paragraph of Section 2.34, “These parameters possess similar physical meaning but differ in the method of calculating vibrational corrections to the values of internuclear distances D and r.m.s. vibrational amplitudes l (Table 1).” D was not present in Table 1 and what is r.m.s.?

Author Response

 We thank every Reviewer for their supportive comments on this manuscript, and for the suggesting changes that we feel have, on incorporation, significantly improved the quality of this work. The most common comments on this work regarded the using of special terms and abbreviations without decoding and clarifications and missing some details of experimental and theoretical methods used. The manuscript has been revised as a consequence including English grammar. Below, giving the responses to Reviewers comments we briefly summarize the major changes made, as it is not possible to succinctly summarize all changes made in this revision. All changes made were given in revised manuscript by yellow color marker.
Nina I. Giricheva, Natalia V. Tverdova, Valery V. Sliznev, Georgiy V. Girichev
Reviewer #2
Reviewer #2 said:
“Sometimes, the authors say “Re2X82-”, sometimes “Re2X8”, what exactly the charges (-2 or 0) of Re2X8 are in their study? And what is the charge in the dissociated ReX4?”
We responded: Two species possessing related stoichiometry Re2X8 were considered in this work – molecule Re2X8 and dianion [Re2X8]2–. The experimental geometrical parameters and the nature of the ReIV-ReIV in Re2X8 molecules (charge=0) existing in the gas phase over rhenium tetrafluoride and the ReIII-ReIII bond in [Re2X8]2– dianions (charge = -2) existing in crystals are
considered and compared.
The charge of the ReF
4 monomer is 0. We have added this information to the text.

Reviewer #2 said: “The authors did not make their work clear to the readers. Why they chose this topic? What impacts are their conclusions? The writing of Abstract and the main text is required to be improved and polished”.
We responded: We have added some details to the abstract and introduction to make it clear that the Re-Re triple bond is a rare phenomenon, and the existence of a rhenium tetrafluoride dimer in the gas phase is a unique fact.

Reviewer #2 said: “It seems the topic of the work is very old, since all the references cited in the work are more than ten years ago. Recent related literature is recommended to be cited and discussed”.
We responded: Following this Comment, we have significantly changed the Introduction and expanded the list of references. Our study is the only direct structural study in which, using the Re2F8 molecule as an example, the existence of the Re-Re triple bond in the gas phase was experimentally shown. In this work, we give a refined structure of the Re2F8 molecule using the latest advances in gas electron diffraction and describe the nature of the chemical bonds in this
molecule in the language of modern quantum chemistry.
Reviewer #2 said: “Some abbreviations appeared in the manuscript for the first time should give the full expression at the same time, such as AO, DFT, QTAIM, LS, HOMO, LUMO, NBO, etc.”
We responded: We have given complete expressions for all notations. All changes in the text of the manuscript are marked in yellow.
Reviewer #2 said: “PBE” is a type of GGA functional, while “PBEPBE” is the keyword of “PBE” functional in Gaussian and some other software.”
We responded: We apologize for our carelessness. In the text, the abbreviation "PBEPBE" has been replaced by "PBE".

Reviewer #2 said: “Authors might provide what are ”RECP-2” and “(F)ECP-3” in their computations.”
We responded: We have seriously revised section 4.1. We introduced clearer notation and shortened the text by removing some details. in Section 4.1 we describe and notate the three variants of functionals (PBE0, B3LYP, PBE) and
basis sets (1,2,3) used in this work. We changed the abbreviation of the quasi-relativistic effective pseudopotential "RECP" to (qRECP) [29], and the abbreviation of the relativistic effective pseudopotential "(F)ECP" to (RECP) [40].
Reviewer #2 said: “There are some grammar issues. For example, “In this work the plates with diffraction patterns of rhenium tetrafluoride [10] were undergo the new photometric experiment using improved equipment and used state-of-art methodics of structural analysis (see Section 4)” in the last paragraph of Introduction, if authors meant to use “undergo”, it should be “underwent”.
We responded: Thank you. The correction was made, and an explanation was added to the Section 4.2 about the new photometric experiment using improved equipment and used state-ofart methodologies of structural analysis.

Reviewer #2 said: “In the first paragraph of Section 2.34, “These parameters possess similar physical meaning but differ in the method of calculating vibrational corrections to the values of internuclear distances D and r.m.s. vibrational amplitudes l (Table 1).” “D” was not present in Table 1 and what is “r.m.s.”?”
We responded: We apologize for our carelessness. This is our oversight, since D= Δr We have excluded this information from the article as it can only be understood by GED researchers. r.m.s. – root mean square. The full expression of this abbreviation is given on page 5.

Reviewer 3 Report

The development of arguments using GED/MS results and DFT orbital calculations is interesting and persuasive. Comparing the Re2F8 molecule and the divalent anion [Re2F8]2-, the authors explain why both dimeric species adopt the eclipse structures and why the Re-Re bond of [Re2F8]2- is shorter than that of Re2F8. The explanations are intriguing.  In these points, this paper may be worthy of publication.  However, it seems to be too specialized and rather difficult to be understood for beginners in this field; eg., QTAIM, NBO, BCP and NPA are used without explanation. At section 4, the calculation procedures should be written in a simpler manner. Following two points shoud be also reconsidered for the publication:

1) At 2.1. Analysis of GED/MS data (Line 65):  Please give some explanation on improvements compared to ref. 10.  Also, please wright more clearly the evidence that Re2F8 was produced.

2) What does mean the asterisk in Table 1?

Author Response

 We thank every Reviewer for their supportive comments on this manuscript, and for the suggesting changes that we feel have, on incorporation, significantly improved the quality of this work. The most common comments on this work regarded the using of special terms and abbreviations without decoding and clarifications and missing some details of experimental and theoretical methods used. The manuscript has been revised as a consequence including English grammar. Below, giving the responses to Reviewers comments we briefly summarize the major changes made, as it is not possible to succinctly summarize all changes made in this revision. All changes made were given in revised manuscript by yellow color marker.
Nina I. Giricheva, Natalia V. Tverdova, Valery V. Sliznev, Georgiy V. Girichev
Reviewer #3
Reviewer #3 said:
“However, it seems to be too specialized and rather difficult to be understood for beginners in this field; eg., QTAIM, NBO, BCP and NPA are used without explanation. At section 4, the calculation procedures should be written in a simpler manner.”
We responded: We have given complete expressions for all notations. We have seriously revised introduction and section 4.1. We introduced clearer notation and shortened the text of section 4.1. by removing some insignificant details.
Reviewer #3 said: Following two points should be also reconsidered for the publication:

Reviewer #3 said: “At 2.1. Analysis of GED/MS data (Line 65): Please give some explanation on improvements compared to ref. 10.” (Now ref. [16]).

We responded: The explanations were added to the Section 4.2. “In Ref. 16 electron diffraction patterns were scanned by their diameter. As a result, the diffraction pattern for each photographic plate was represented by a set of 301 points. In this case, the complete statistical sample for all electron diffraction patterns was 4214 points. In contrast of [16] we significantly increased the sample of scanning results and precision of optical density measurements. To achieve this goal, we used a modified MD-100 microdensitometer [50] and measured the optical density on each plate by scanning an area of 10×130 mm with a step of 0.1 mm along 33 equidistant scanning lines. As a result of new
microphotometric measurements, the statistical sample of primary experimental data was significantly increased in comparison with [16] and amounted to more than 500,000 points.” In addition, to obtain starting approximations to the vibrational characteristics of the molecule, we used the force field obtained by the DFT method instead of the estimated force constants in [16] (Section 4.2).
Reviewer #3 said: “Also, please wright more clearly the evidence that Re2F8 was produced.”
We responded: Using the GED/MS method gives confidence that the gas phase contains the molecular form Re2F8. The [ReF3]+(100), [ReF4]+(~50), [Re2F7]+(~50) ions have the highest intensity in mass-spectra. These ions are the result of dissociative ionization of Re2F8 molecules under electron impact.
Reviewer #3 said: “What does mean the asterisk in Table 1?”
We responded: We apologize for our carelessness. This is mistake. We removed an asterisk. 

Round 2

Reviewer 2 Report

Typically, authors prepared the Abstract in one paragraph.

"d-AOs" in the Abstract should be combined with its full expression.

They updated the references, still a little surprising, the latest one was published in 2016. 

Author Response

We are deeply grateful to the Reviewers for their very attentive attitude to this manuscript. Below are responses to comments from Reviewers.

Reviewer #2

Comment 1:

Typically, authors prepared the Abstract in one paragraph.

Answer:

Thank you. We took this note into account and edited the Abstract.

Comment 2:

"d-AOs" in the Abstract should be combined with its full expression.

Answer:

Thank you. "d-AOs" changed to "d-atomic orbitals".

Comment 3:

They updated the references, still a little surprising, the latest one was published in 2016. 

Answer:

The Re2F8 dimer is a unique example of the existence of a molecule with a metal-metal triple bond in the gas phase, the structure of which was established in our work experimentally by a direct structural method, and the features of the electronic structure were studied by calculation. We hope that this work will revive interest in numerous compounds with metal-metal bonds, since at present there is a powerful tool - quantum chemical calculations, which make it possible to describe in detail the nature of this bond and evaluate its strength.

Reviewer 3 Report

The manuscript has been well revised and is ready for publication. Please check the following points again.

1) Line 47: low → high (?)

2) The caption of Fig. 2 should be written below the figure.

3) In Table 3, remove the asterisk.

4) Line 471: [Sipachev] → [16] (?)

Author Response

We are deeply grateful to the Reviewers for their very attentive attitude to this manuscript. Below are responses to comments from Reviewers.

Reviewer #3

Comment 1:

Line 47: low → high (?)

Answer:

For clarity, we replaced the phrase «As noted in the literature, in contrast to their related compounds with a low degree of oxidation, ReIV (d3 configuration) complexes…», in new redaction with “As noted in the literature, in contrast to their related compounds (ReIII, d4 configuration) with a lower degree of oxidation, the ReIV (d3 configuration) complexes are not prone to the formation of metal-metal bonds [1].”

Comment 2:

The caption of Fig. 2 should be written below the figure.

Answer:

Thank you. We fixed this error.

Comment 3:

In Table 3, remove the asterisk.

Answer:

Thank you. This is our mistake. Instead of an asterisk, there should be a superscript a). We have corrected this error.

Comment 4:

Line 471: [Sipachev] → [16] (?)

Answer:

Thank you. We changed the reference [Sipachev] to [19-21].
